# Only Small Effects of Mindfulness-Based Interventions on Biomarker Levels of Inflammation and Stress: A Preregistered Systematic Review and Two Three-Level Meta-Analyses

**DOI:** 10.3390/ijms24054445

**Published:** 2023-02-23

**Authors:** Jakob Grasmann, Frederick Almenräder, Martin Voracek, Ulrich S. Tran

**Affiliations:** Department of Cognition, Emotion, and Methods in Psychology, Faculty of Psychology, University of Vienna, Liebiggasse 5, A-1010 Vienna, Austria

**Keywords:** MBIs, mindfulness, biomarkers of inflammation and stress, interleukin, C-reactive protein, cortisol, three-level meta-analysis

## Abstract

Mindfulness-based interventions (MBIs) have a positive effect on biomarkers of inflammation and stress in patients with psychiatric disorders and physical illnesses. Regarding subclinical populations, results are less clear. The present meta-analysis addressed the effects of MBIs on biomarkers in psychiatric populations and among healthy, stressed, and at-risk populations. All available biomarker data were investigated with a comprehensive approach, using two three-level meta-analyses. Pre–post changes in biomarker levels within treatment groups (*k* = 40 studies, total *N* = 1441) and treatment effects compared to control group effects, using only RCT data (*k* = 32, total *N* = 2880), were of similar magnitude, Hedges *g* = −0.15 (95% *CI* = [−0.23, −0.06], *p* < 0.001) and *g* = −0.11 (95% *CI* = [−0.23, 0.001], *p* = 0.053). Effects increased in magnitude when including available follow-up data but did not differ between type of sample, MBI, biomarker, and control group or duration of the MBI. This suggests that MBIs may ameliorate biomarker levels in both psychiatric and subclinical populations to a small extent. However, low study quality and evidence of publication bias may have impacted on the results. More large and preregistered studies are still needed in this field of research.

## 1. Introduction

Mindfulness is defined as “the awareness that emerges through paying attention on purpose, in the present moment, and nonjudgmentally to the unfolding of experience moment by moment” [1] (p. 145). The construct of mindfulness has its roots in Eastern spiritual traditions and can be seen as a fundamental core of Buddhist meditation [2]. Mindfulness was first introduced as a therapeutic approach by Kabat-Zinn in the 1970s and led to the development of mindfulness-based stress reduction (MBSR), originally designed as an eight-week treatment protocol for patients with chronic pain [3,4].

Other standardized mindfulness-based interventions (MBIs) besides MBSR are, for example, mindfulness-based cognitive therapy (MBCT [5]) and mindfulness-based relapse prevention (MBRP [6]). Further, currently there exist several adaptions that are based on the established MBSR program. Examples include the health enhancement through mindfulness (HEM) program, mindful awareness practices (MAPs), brief mindfulness meditation programs (MM), or MBIs specifically adapted for workplace and university settings (for an overview, see Appendix A). Multiple meta-analyses have provided positive results regarding the efficacy of MBIs for the treatment of different psychiatric symptoms and disorders. Accordingly, MBIs represent an effective treatment option for patients with depression, anxiety disorders, substance use disorders, sleep disorders, and somatoform disorder [7,8,9,10,11,12].

Current research provides also evidence for links between neuroinflammation and physical and mental health [13,14]. In the field of mental illness, elevated levels of certain biomarkers, such as C-reactive proteins [15], neuropeptides [16], cytokines [17], and chemokines [18], are associated with various psychiatric symptoms (see also [19]). The direction of association may differ for other biomarkers (for details and comprehensive overview, see Table 1).

For all biomarkers of inflammation and the stress-related biomarkers cortisol, alpha-amylase (AA), and adrenocorticotropic hormone (ACTH), elevated levels indicate worsening health status [18,20,21,22,24,25,26,37,38]. For the cortisol-antagonist dehydroepiandrosterone sulfate decreased levels indicate worsening health status (DHEA-S [23]). Of the other biomarkers (Table 1), increased DNA methylation of the serotonin transporter gene *SLC6A4* (*SLC6A4* DNAm) seems to have a negative effect on mood and the response to stress [35], whereas reductions of leukocyte telomere length (LTL), methylation of the binding protein FKBP5 (FKBP5m), brain-derived neurotrophic factor (BDNF), neuropeptide-Y (NP-Y), oxytocin (sOXT), and of the epidermal growth factor (EGF) appear to be associated with worsening health status [30,31,32,34,36]. For example, BDNF levels are decreased in Alzheimer patients [30], and a reduction of LTL is prevalent in patients with depression [33].

There is meta-analytic evidence that MBIs not only lead to improvements in psychiatric symptoms and mental health, but also in biomarkers of inflammation and stress in psychiatric patients [39] and in patients with various physical illnesses, for example, cancer or HIV [40,41]. Further, MBIs may also have similar beneficial effects on biomarkers of inflammation and stress among healthy persons, stressed persons (e.g., individuals in highly stressful work or with heightened levels of self-reported stress), and persons at-risk (e.g., individuals in circumstances that put them at a heightened risk for the development of stress-related and mental illnesses) [42,43,44]. However, the results of these primary studies were overall ambiguous, and, where reported, positive effects were of only small size, probably moderated by stress load (i.e., small effects among lowly stressed individuals, but larger effects among more stressed individuals [45]). Further, available meta-analyses and reviews have pointed out that treatment effects did not last until the time of follow-up [39,41] and that other psychological interventions, such as cognitive behavioral therapy (CBT), may have similar effects as well [41].

However, the current status of meta-analytic knowledge on this topic may need improvement. First, most primary studies in this field of research reported multiple biomarkers. Previous meta-analyses selectively aggregated only part of this evidence, by picking only one biomarker per study or by aggregating biomarkers in separate analyses. This approach also aimed to deal with the methodological problem that effect sizes need to be independent in the classic meta-analytic approach [46]. However, newer approaches, such as three-level meta-analysis (TLMA; e.g., [47]), can handle dependent effect sizes and are able to aggregate all the available evidence that is contained in primary studies in a single analysis. TLMA does so by partitioning the variation in the effect sizes on three levels, namely, within participants (Level 1), within studies (Level 2), and between studies (Level 3). None of the extant meta-analyses in this field [39,40,41] has made use of TLMA or other, related methods.

Second, previous meta-analyses have investigated primarily studies with patients with psychiatric disorders or physical illnesses, thereby mixing effects, which may not relate to mental health only. Neither [39], nor [40] have included studies with healthy, stressed, or at-risk participants. The meta-analysis of O’Toole et al. [41] included three such studies (out of 19) but did not investigate possible differences to psychiatric patients. Thus, it is currently unclear whether the effects of MBIs on biomarkers of inflammation and stress extend from psychiatric to subclinical populations. The currently available evidence on such populations still awaits appropriate meta-analytic aggregation.

The current meta-analysis provides a comprehensive investigation of the effects of MBIs on biomarkers related to inflammation and stress in both psychiatric and subclinical (healthy, stressed, and at-risk) populations but excluded patients with physical illnesses. Two meta-analyses were conducted, using all biomarkers within each study. The first meta-analysis (TLMA 1) investigated the effects of MBIs within treatment groups to provide a raw estimate of the possible magnitude of treatment effects. We drew on all available evidence, utilizing also data of single-arm studies, in addition to randomised and non-randomised trials (RCTs and non-RCTs). The second meta-analysis (TLMA 2) focused on the comparison of treatment with control groups, using only RCT data. We thereby also addressed anew the question of whether psychological interventions other than MBIs might have similar effects on biomarkers [41].

The current meta-analysis thus beneficially extends extant meta-analyses and provides novel insights. Broadening the range of eligible studies from psychiatric to subclinical populations, drawing on different study designs, and using all available biomarker data within each study were methods aimed at a more comprehensive evaluation of the effects of MBIs on biomarkers than previous meta-analyses, increasing overall sample size and thus analytic power. This larger corpus of primary studies was also intended to enable more detailed subgroup analyses, concerning sample characteristics, MBIs, duration of MBIs, biomarkers, and risk of bias. It was thereby planned to investigate if MBIs had a stronger effect on biomarkers in psychiatric, healthy, stressed, or at-risk populations, or if the effects of MBIs were more pronounced for certain types of biomarkers than for others. The second meta-analysis (TLMA 2) aimed to control for threats to internal validity and to enable comparisons of MBIs with other active treatments. It thus provides information on whether the effects of MBIs on biomarkers are specific for them or may be more general, i.e., valid for other psychological treatments as well. Lastly, the focus on effects related to mental health only aimed to rule out confounds that could be due to physical illnesses.

## 2. Materials and Methods

A protocol for this study was preregistered at OSF (https://osf.io/8h53c (registered on 27 March 2021)). The meta-analytic workflow followed the PRISMA 2020 checklist [48].

### 2.1. Study Eligibility Criteria

Inclusion and exclusion criteria are provided in detail in Appendix A. Studies were eligible if they examined the effects of mindfulness-based interventions (MBSR, MBCT, MBRP, or any other MBI) on biomarkers related to processes of inflammation and stress (see Table 1) in samples of healthy, stressed, or at-risk persons or patients with psychiatric diagnoses. Studies with patients with physical illnesses were not eligible. Studies needed to provide quantitative data (pre- and post-test scores) from a prospective research design (randomised controlled trial, non-randomised controlled trial, single-arm, or multiple-arm trial). Studies with qualitative designs were excluded. Lastly, full texts needed to be available, i.e., studies either needed to be published in a peer-reviewed journal or as grey literature. Reviews, case reports, and dissertation abstracts were excluded.

### 2.2. Literature Search Strategy

The literature search was based on the electronic databases PsycInfo, Scopus, Web of Science, PubMed, and Google Scholar. The final search was completed on 8 June 2021. Using Boolean operators, a search string was created by combining different search terms for mindfulness-based interventions and biomarkers of inflammation and stress (see Appendix A). No limit was defined for the year of publication, and no language restrictions were set. Where necessary, the structure of the search string was adapted to fit the search engine of the specific database.

### 2.3. Risk of Bias Assessment

For the risk of bias assessment, the Cochrane risk-of-bias tool for randomized trials (RoB 2) was used [49]. This tool consists of five domains, each assessed with multiple items: randomisation process, deviations from intended interventions, missing outcome data, measurement of the outcome, and selection of reported results. Overall risk of bias was determined based on the domain bias ratings. For single-arm studies, only the domains missing outcome data, measurement of the outcome, and selection of reported results were rated.

### 2.4. Effect Sizes for Study Outcomes

Two separate TLMAs were conducted. TLMA 1 aimed at investigating the effects of MBIs on biomarkers within the treatment group. For this analysis, effect sizes in the metric of Cohen *d* were calculated, using Formulae (1) and (2) [50] (pp. 28–29) and the tool of Lakens [51].
(1)$$d=\frac{M_{post\;}–\;M_{pre\;}\;}{SD_{diff}}\times \sqrt{2\left(1-r\right)}$$
(2)$$SE=\sqrt{\left(\frac{1}{n}+\frac{d^{2}}{2n}\right)2\left(1-r\right)}$$

*M_post_* and *M_pre_* denote pre-test and post-test mean scores. *SD_diff_*, the *SD* of the pre-test and post-test differences, was estimated from the pre- and post-*SD*s, if not directly provided (the exact calculation is described in [51], Formula (8)). Obtained effect sizes were transformed into Hedges *g*, using the correction factor *J* (Formulae (3) and (4); [50] (p. 27)) that adjusts for the small-sample bias of Cohen *d*.
(3)$$J=1-\frac{3}{4df-1}$$
(4)g=d×J

The effect sizes in (1) and (4) were adjusted for the correlation between the pre-test and post-test scores. As this correlation was not reported in primary studies, we used three different values (*r* = 0.1, 0.5, and 0.9) for calculations, with *r* = 0.5 being used in the main analysis, whereas the other two values were in sensitivity analyses.

TLMA 2 compared the effects of MBIs on biomarkers between treatment and control groups. Effect sizes in the metric of Hedges *g* were calculated with the *escalc* function of the metafor package (see Section 2.7), using Formula (5):(5)$$g=J_{T}\left(\frac{M_{post,T}-M_{pre,T}}{SD_{pre,T}}\right)-J_{C}\left(\frac{M_{post,C}-M_{pre,C}}{SD_{pre,C}}\right)$$

This formula is based on the pre-test and post-test mean scores of treatment and control groups; *J_T_* and *J_C_* relate to the correction factor *J* in the treatment and control groups, respectively. For the *SE* of this effect size, Formula (16) in [52] was used (slightly adapted, as explained in https://www.metafor-project.org/doku.php/analyses:morris2008#computing_the_difference_in_the_standardized_mean_change (accessed on 1 June 2021)).

For primary studies that did not report sufficient information to directly apply the above formulae, alternative methods (see [53] (p. 216)) were used to calculate the required effect sizes. For studies that reported a median and its range, formulae by [54,55] were applied to estimate the corresponding mean and *SD*. If studies reported only an effect size and if no other outcome data were available, values were transformed into the above effect size metrics, where applicable (see Appendix A).

The direction of the obtained effect sizes was aligned with the direction of beneficial levels of the biomarker in question (see Section 1 and Table 1). In most cases, decreases in biomarker levels were beneficial, while for some biomarkers (e.g., sOXT) increases were favourable. In these cases, the sign of the effect size was switched to allow for a meaningful interpretation of the aggregated effect size. A negative sign therefore indicates beneficial effects.

### 2.5. Effect Moderators

In total, six variables were used as moderators for planned subgroup analyses, the first five of which were investigated in both TLMA 1 and TLMA 2, whereas the sixth was only investigated in TLMA 2. First, sample characteristics were investigated, contrasting healthy, stressed, and at-risk samples and samples with patients with a diagnosis of mental illness. The ‘stressed’ category included studies with participants in highly stressful work, study, or sport-related situations; participants with heightened levels on the Perceived Stress Scale [56]; or participants performing the Trier Social Stress Test [57]. The ‘at-risk’ category included studies with participants in circumstances that put them at a heightened risk for the development of stress-related and mental illnesses, e.g., dementia caregivers, older adults, or people living in poor living conditions.

The second and third moderator variables were type and duration of MBI. We differentiated three categories. The first category included the established eight-week programs MBSR, MBCT, and MBRP; the second category comprised adaptations of MBSR (see Appendix A); the third category included acceptance and commitment therapy (ACT [58]), which is considered a “mindfulness-informed” intervention [59]. It places less emphasis on formal meditation than the interventions in the first two categories and, hence, may differ from them. The duration of MBI was coded as less than eight weeks (<8 weeks), eight weeks (8 weeks), and more than eight weeks (>8 weeks). This coding scheme followed the convention that the most established, standardised MBIs, such as MBSR or MBCT, are designed for a period of 8 weeks.

Type of biomarker (see Table 1 details) was the fourth moderator variable, differentiating three categories, based on their physiological properties: biomarkers related to stress (AA, ACTH, cortisol, and DHEA-S), biomarkers related to inflammation (CRP, IL-6, IL-8, IL-10, IL-1β, IL-1ra, TNF-α, and NF-κB), and biomarkers only indirectly related to inflammation and stress (BDNF, EGF, FKBP5m, LTL, NP-Y, *SLC6A4* DNAm, and sOXT).

Fifth, moderating effects of study quality were investigated, using the overall RoB 2 rating [49]. We differentiated for low risk, some concerns, and high risk of bias for individual studies. In TLMA 2, we specifically investigated, as a sixth moderator, type of control group. We differentiated between active control groups, wherein participants attended any other type of active intervention (e.g., health education programs or cognitive-behavioural therapy [CBT]), and passive control groups, which did not involve any active form of intervention, for example, waiting lists or no treatment at all. Treatment-as-usual (TAU) conditions were coded as active, as previous meta-analyses indicated large heterogeneity in TAU conditions, with some of them qualifying as active interventions ([60]; but see [61]).

### 2.6. Coding Procedure and Intercoder Reliability

Coding was conducted independently by two authors (JG and FA). To assess the intercoder reliability, Brennan and Prediger’s κ (KBP) [62] was used for categorial variables and intraclass correlations (*ICC*; two-way random effects model, absolute agreement) for metric variables, using the irrCAC and the irr R packages for calculation [63]. KBP values were between 0.79 and 1, indicating substantial to perfect agreement [62]. *ICC* values varied between 0.86 and 1, which corresponds to good-to-excellent agreement as well [64]. Any disagreements were resolved through discussion and consensus.

### 2.7. Statistical Analyses

The two TLMAs were performed in R (Version 4.0.1), using the metafor package [65]. TLMA is a random-effects model that deals with the interdependency of effect sizes originating from the same study by partitioning the variation in effect sizes between participants (Level 1; sampling error), outcomes (Level 2; within studies), and studies themselves (Level 3) [47]. For parameters estimation, restricted maximum likelihood estimation (REML) was applied. For the estimation of standard errors, *p* values, and 95% confidence intervals (*CI*s), Knapp and Hartung’s [66] adjustment was applied. For the assessment of heterogeneity, the *I*^2^ statistic was used, which describes the amount of variation due to true effect-size heterogeneity, i.e., beyond mere sampling error (low heterogeneity = 25%, moderate heterogeneity = 50%, and high heterogeneity = 75% [67]). We also report the standard deviation τ of the estimated heterogeneity of effect sizes on Levels 1 and 2.

### 2.8. Outcome Visualisation

For the visualisation of individual effect sizes and the results of the data synthesis, R functions specifically designed for TLMAs [68] were applied. We present three-level forest plots, three-level funnel plots, and caterpillar plots of all effect sizes.

### 2.9. Publication Bias

To assess the risk of publication bias, we visually inspected three-level funnel plots [68] first. Second, the three-level Egger’s regression test [69,70] and the Begg-Mazumdar rank-order correlation test [70,71] were applied. Further, *p*-uniform* was used [72]. The goal of *p*-uniform* is to provide a pooled effect estimate corrected for possible publication bias, while also testing for between-study variance. This method is not directly applicable to the dependent effect sizes in TLMA but requires independent effect sizes. Thus, one effect size per study was randomly drawn.

### 2.10. Outlier Analyses

For the identification of potential outliers, first, the *find.outliers* function of the dmetar R package was applied [73]. This function classifies outliers based on the overlap of the confidence intervals of the primary study effect size and the pooled effect size estimates. If the two confidence intervals do not overlap, the primary study effect size should be excluded [73]. Second, as stated in the preregistered analysis plan, we also aimed to use the graphic display of heterogeneity (GOSH) plot [74,75]. For the GOSH plot, the same meta-analytic model is fitted to all possible subsets of the included primary studies. However, the GOSH diagnostic plots were ultimately not used. The tool is currently not adapted to the dependent data structures in TLMA and could potentially lead to ambiguous results that would not lend themselves to clear-cut interpretation.

## 3. Results

Figure 1 presents the PRISMA flow diagram for the two meta-analyses. In total, *k* = 45 studies, with 47 independent samples, were included in the two TLMAs, with a total sample size of *N* = 3140 participants. Mean participant age was 37.10 years; 60% of participants were women. Thirteen (29%) studies investigated healthy participants; 14 investigated (31%) stressed participants; three (7%) investigated at-risk participants; and 15 (33%) investigated patients with psychiatric diagnoses. Regarding MBI, 12 studies (27%) used MBSR, MBCT, or MBRP; 32 studies (71%) adapted MBI programs; and one study (2%) used ACT. Thirty-five studies included a control group, of which 18 (51%) were active controls, whereas 17 (49%) were passive controls. Active controls comprised only in two (11%) studies CBT; otherwise, interventions such as cognitive control trainings, health enhancement programs, or relaxation trainings were used. Thirty-one of the 35 studies with a control group were RCTs; four were nonrandomized [76,77,78,79]. Seven studies (16%) conducted a follow-up assessment. Important characteristics of all included studies are presented in Table 2.

The current meta-analyses included all studies from the prior related meta-analyses [41] (*k* = 3) and [39] (*k* = 7) that matched the present inclusion criteria. Compared to these two previous syntheses, the current meta-analysis included 35 additional studies and added 68 effect sizes and 2628 participants.

### 3.1. Risk of Bias

There were some concerns for a total of 34 (75%) studies (see Figure 2). In most of these cases (*k* = 22; 65%), concerns arose from possibly selected outcomes; only a small number of studies had been preregistered. In four studies (9%), there was high risk of bias, while only seven studies had a low risk of bias (16%). Studies with high risk of bias either had problems with the randomisation process (*k* = 1; 25%) or missing outcome data (*k* = 3; 75%). Overall risk of bias ratings was used for the subgroup analyses (see below).

### 3.2. Pooled Effect Estimates in the Two TLMAs

TLMA 1, which investigated the effect sizes of interventional effects within treatment groups only, comprised *k* = 40 studies with 42 independent samples and 91 effect sizes. The total sample size was *n* = 1441. TLMA 2, which investigated effects between intervention and control groups and comprised *k* = 33 studies with 35 independent samples and 79 effect sizes. Here, the total sample size was *n* = 2880. Figure 3 shows caterpillar plots with all individual effect sizes for the two TLMAs. Figure 4 and Figure 5 present three-level forest plots of the effect size distribution aggregated on the study level (Level 3).

The pooled effect estimate in TLMA 1 was *g* = −0.15 (*SE* = 0.04), 95% *CI =* [−0.23, −0.06], showing that MBIs beneficially affected biomarkers of inflammation and stress within the treatment groups (see Table 3). Heterogeneity and total variance, not attributable to sampling error, were similar for Levels 2 and 3.

The pooled effect estimate in TLMA 2, *g* = −0.11 (*SE* = 0.06), 95% *CI* = [−0.23, 0.001], was similar in magnitude to that of TLMA 1 but narrowly missed nominal significance (see Table 3). Heterogeneity and total variance, not attributable to sampling error, were larger for Level 2 than Level 3; i.e., more variation was attributable to differences within studies (biomarkers) than between studies.

The within-study (Level 2) effect size variance was significant for both TLMAs (likelihood ratio tests; *p*s < 0.001), but the between-study (Level 3) variance was significant only for TLMA 1 (χ^2^(1) = 6.92, *p* = 0.009; TLMA 2: χ^2^(1) = 1.47, *p* = 0.23). This indicated that the between-study effect size variance appeared negligible for TLMA 2. However, the three-level structure was still kept for further analysis.

### 3.3. Sensitivity Analyses

For TLMA 1, using *r* = 0.1 and *r* = 0.9 instead of *r* = 0.5 for the calculation of the individual effect sizes, the pooled effect size was *g* = −0.14 (*SE* = 0.04), 95% *CI* = [−0.22, −0.05], *p* = 0.002 and *g* = −0.15 (*SE* = 0.04), 95% *CI* = [−0.24, −0.06], *p* < 0.001 for TLMA 2, respectively. The similar point estimates and the overlap of the confidence intervals indicated that the choice of correlation between pre- and post-test values did not lead to substantially different pooled effect estimates. A correlation of *r* = 0.5 was thus kept for all further analyses.

The outlier analysis identified individual effects sizes of six studies [17,42,80,97,103,106] as outliers in TLMA 1, and individual effect sizes of three studies of [76,97,100] as outliers in TLMA 2. Excluding the outlying data points, the pooled effect estimate was somewhat diminished in magnitude for TLMA 1 but increased for TLMA 2, providing now a nominally significant overall effect as well (Table 3).

### 3.4. Follow-Up Data

For a list of studies that reported follow-up effects, see Appendix A. There were five studies (#ES = 16) reporting follow-up data for TLMA 1 and four studies (#ES = 14) for TLMA 2. The mean length of the follow-up periods was 18.8 weeks (range: 8 to 36 weeks). Pooled effect estimates were *g* = −0.08 (*SE* = 0.14), 95% *CI* = [−0.37, 0.21] for TLMA 1 and *g* = −0.15 (*SE* = 0.14), 95% *CI* = [−0.45, 0.15] for TLMA 2, indicating nominally not significant effects at follow-up. However, the magnitude of these pooled effects was similar to the magnitude of the pooled effects at post-treatment (Section 3.1). Analyzing post-treatment and follow-up effects in the same models, there were no differences (moderator analysis; *p* = 0.55 for TLMA 1 and *p* = 0.60 for TLMA 2). Including follow-up data slightly increased the magnitude of the pooled effect estimate in TLMA 1, *g* = −0.17 (*SE* = 0.04), 95% *CI* = [−0.23, −0.06], and TLMA 2, *g* = −0.15 (*SE* = 0.05), 95% *CI* = [−0.25, −0.05], rendering the pooled effect now also nominally significant (see Table 3).

### 3.5. Publication Bias

Publication bias was checked via multiple methods. Figure 6 shows three-level funnel plots for both TLMAs. The visual inspection of the two funnel plots did not indicate strong deviations from a symmetrical distribution of the effect sizes. However, the three-level Egger’s and Begg–Mazumdar’s correlation tests suggested effect size asymmetry in TLMA 1 (*p* = 0.011 and *p* < 0.001; TLMA 2: *p* = 0.45 and *p* = 0.37). This indicated that smaller and less precise studies reported larger beneficial effects than larger and more precise studies.

The results of the *p*-uniform* analyses were not suggestive of publication bias (test for publication bias: *p* = 0.94 in TLMA 1, *p* = 0.82 in TLMA 2). Corrected overall effect estimates were *g* = −0.17, 95% *CI* = [−0.30, −0.04], *p* = 0.009 (TLMA 1) and *g* = −0.20, 95% *CI* = [−0.33, −0.07], *p* = 0.035 (TLMA 2). However, these results need to be interpreted with caution, as they were based on a random sample of effect sizes (one per study only; #ES = 40 and 35, respectively, in the two analyses).

### 3.6. Subgroup Analyses

Subgroup analyses were performed for type of sample, type of MBI, duration of MBI, type of biomarker, and risk of bias in TLMA 1 and TLMA 2. Type of control group was investigated as a further moderator in TLMA 2. As there was only a small number of studies with at-risk participants, studies with stressed and at-risk populations were thus combined into one category. There were no significant differences between any of the categories in either of the two TLMAs (all *p*s > 0.05; see Table 4 and Table 5). However, descriptively, effects of MBIs appeared to be more relevant for the stress-related and indirect biomarkers in TLMA 1 and 2 and less relevant for the inflammation-related biomarkers. Additionally, effects appeared to increase with risk of bias in both TLMAs; among the small number of studies with no risk of bias, there seemed to be no relevant effects.

## 4. Discussion

This systematic review and meta-analysis examined the effects of MBIs on a broad variety of biomarkers related to inflammation and stress in samples from both psychiatric and subclinical populations, excluding samples with physical illnesses. In analyses controlling for outliers or incorporating available follow-up data, we obtained meta-analytic evidence that MBIs decreased biomarker levels by a small margin (of the order of *d*~−0.15, rounding to the nearest increment of 0.05), both in pre–post changes within treatment groups and when comparing treatment groups with active or passive control groups in RCT data.

The present results are in line with previous findings [39] and comparable to results from a recent meta-analysis [40], which reported a reduction of biomarkers of inflammation ranging from *d* = −0.14 for CRP to *d* = −0.35 for IL-6 in patients with physical illnesses. Sanada et al. [39] synthesized data of five primary studies with psychiatric patients, all of which were included also in the present meta-analysis. The present results thus extend on these previous findings and suggest that the effects of MBIs on biomarkers generalise also to the subclinical range.

O’Toole et al. [41] reported no differences between MBIs and CBT and that effects did not last until the last follow-up. We found that the effects of MBIs exceeded those of the control groups and remained until follow-up. Synthesising data of 33 RCTs in TLMA 2 (vs. 19 in [41]), the present meta-analysis had higher analytic power, used all available data within studies, and further did not mix psychiatric and physical diagnoses. However, the present results still have to be interpreted with caution. Only data of seven studies were available at follow-up and only two studies compared MBIs to CBT. More studies are clearly still needed to draw firmer conclusions.

Further, none of the moderator variables appeared to moderate the treatment effects in either of the two TLMAs. Thus, contrary to expectation, there were no detectable differences between populations, MBIs, and biomarkers of inflammation and stress. Nevertheless, descriptively, there was some indication that MBIs could affect inflammation-related biomarkers less than stress-related and indirect biomarkers. Puhlmann et al. [45] reported reductions specifically in inflammation-related biomarkers in vulnerable persons. This finding may need replication, but more data are generally needed, concerning the possible moderators of treatment effects. Conspicuously, effects were also smaller in studies with low risk of bias than in studies with higher risk of bias. Effects of publication bias were further likely in TLMA 1. Study quality is a pervasive problem in MBI research [7,116,117]. There is a need to address these obvious shortcomings in future research through study preregistration and open data.

### 4.1. Limitations

The two TLMAs have some limitations. First, the sample size of the included studies varied between nine and 281 participants, with a total of nine studies having sample sizes *n* < 20. In light of the overall small magnitude of effects, small samples also decrease meta-analytic power. Second, the included studies reported a large variety of biomarkers, with some biomarkers being only reported once. Therefore, biomarkers had to be clustered into groups, which did not allow obtaining biomarker-specific effect size estimates. Third, for many of the studies, there were at least some concerns regarding risk of bias. In most cases, the concerns were based on the possibility of selective reporting, which cannot be ruled out for studies, which were not preregistered. Even though study quality did not appear to significantly moderate the effects, pooled estimates were smallest in low-risk studies, whereas they were largest in high-risk studies. This means that the present meta-analyses might have overestimated the overall effect. Lastly, there are not many options for the assessment of publication bias, other than Egger’s and Begg–Mazumdar’s tests [70], currently available for TLMA. More advanced, recent approaches, such as, for example, *p*-uniform* [72], could only be applied to subsets of the effect sizes.

### 4.2. Future Research

Future research investigating the effects of MBIs on different biomarkers should consider the following aspects. First, larger sample sizes are needed in this field of research to detect small effects with higher analytic power and precision. Second, primary studies should be preregistered and provide open data to minimise the risk of bias. Third, more RCTs are required for the biomarkers AA, DHEA-S, IL-8, IP-10, IL-1β, IL-1ra, NF-κB, *SLC6A4* DNAm, LTL, FKBP5m, BDNF, NP-Y, sOXT, and EGF (see Table 1 details) to allow for biomarker-specific effect estimates. Since all biomarkers are measured via blood or saliva samples, it could be possible to examine all of these biomarkers from the same blood or saliva sample. Promising biomarkers besides the most established ones (CRP, IL-6, TNF-α) appear to be IL-8, NF-κB, LTL, EGF, and *SLC6A4* DNAm, considering reported effect sizes. Biomarkers could be further informative as quantitative measures of treatment success in mediation studies on the treatment efficacy of MBIs [117]. Alternative and less subjective measures than self-reported mental health, such as provided by biomarkers, are needed in this field of research.

## 5. Conclusions

This meta-analysis suggests that MBIs have a positive effect on various biomarkers related to inflammation and stress both in psychiatric as well as subclinical populations. However, effects were of comparatively small magnitude, and other active psychological interventions besides MBIs (e.g., CBT) still might lead to comparable results. Problems of study quality need to be addressed in this field of research. More large and preregistered studies, along with data for some specific biomarkers, are still needed.

## Figures and Tables

**Figure 1 ijms-24-04445-f001:**
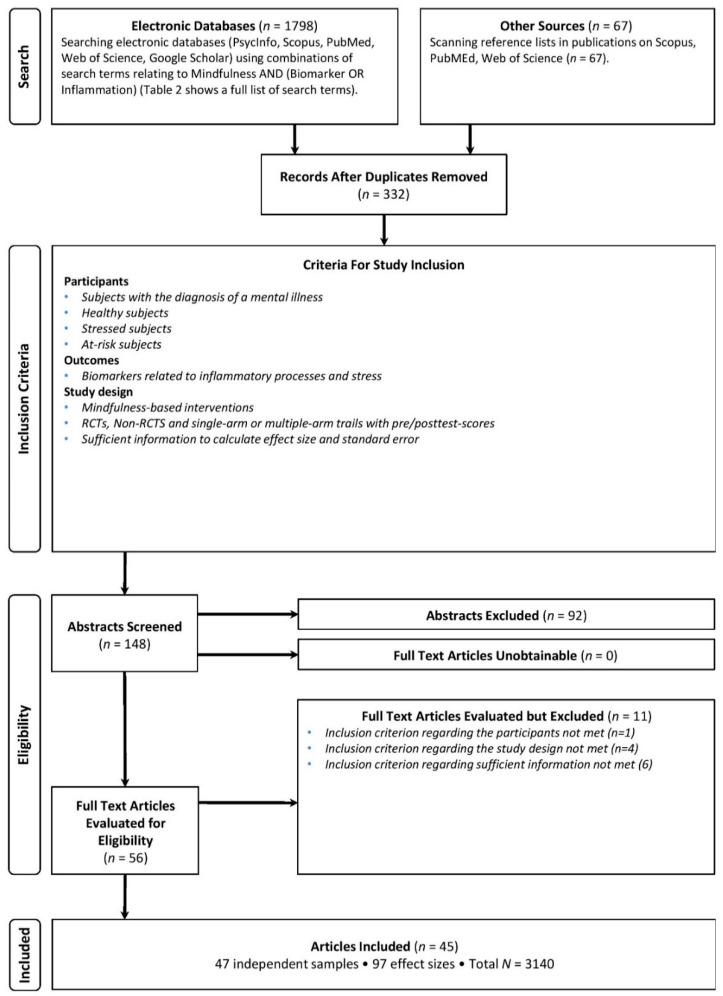
PRISMA Flow Diagram.

**Figure 2 ijms-24-04445-f002:**
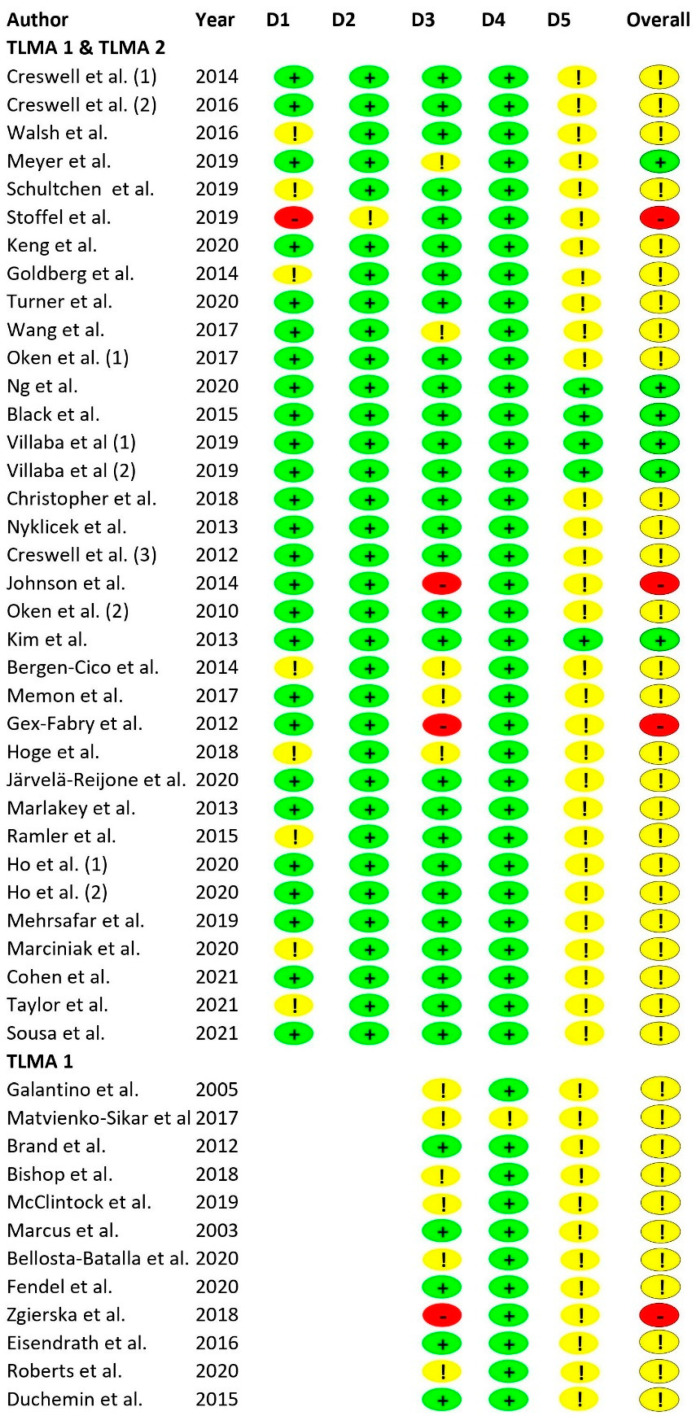
Results of the Risk of Bias Assessment (RoB 2). Note. References of included studies are provided in Table 2. Numbers in parentheses refer to different samples of the respective studies (see Table 2 for details). TLMA = three-level meta-analysis; + = low risk; ! = some concerns; - = high risk; D1 = randomization process; D2 = deviations from intended interventions; D3 = missing outcome data; D4 = measurement of the outcome; D5 = selection of reported result.

**Figure 3 ijms-24-04445-f003:**
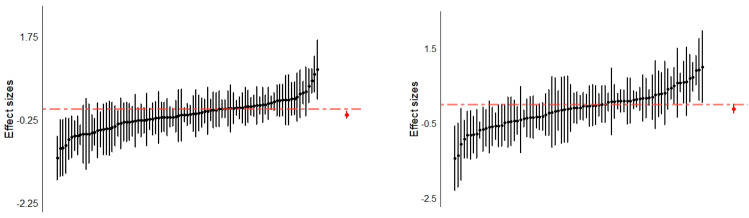
Caterpillar Plots of Individual Effect Sizes in TLMA 1 (**Left**) and TLMA 2 (**Right**). Note. The black dots represent all biomarker effect sizes (Level 2) and their corresponding 95% confidence intervals. The red dot represents the pooled effect estimate, the red line the absence of an effect (*g* = 0).

**Figure 4 ijms-24-04445-f004:**
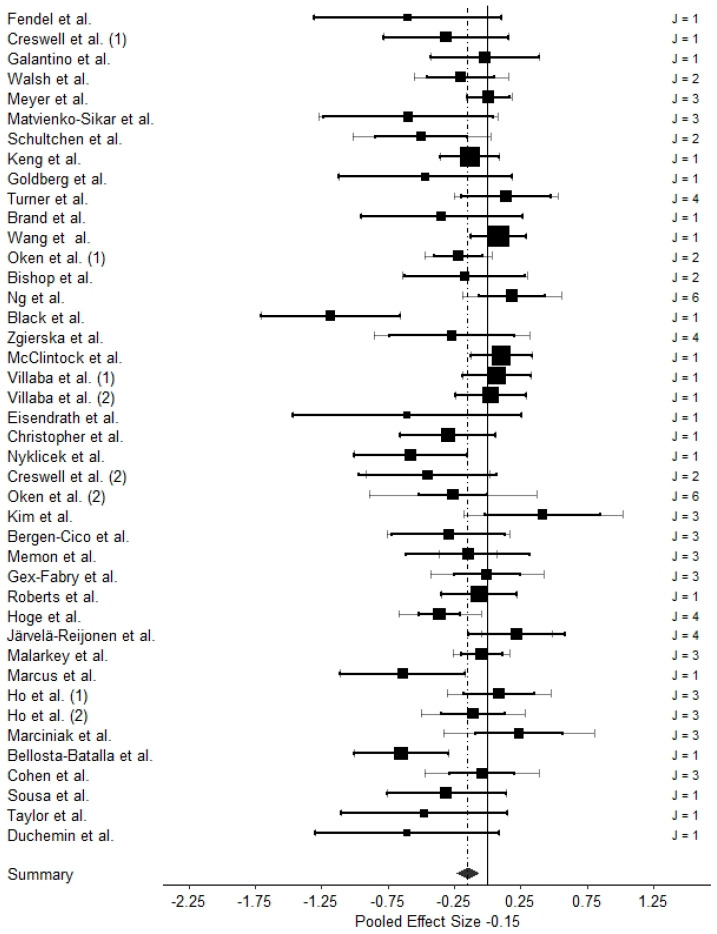
Three-Level Forest Plot of TLMA 1. Note. References of included studies are provided in Table 2. Numbers in parentheses refer to different samples of the respective studies (see Table 2 for details). Aggregated effect sizes on the study level (Level 3) are represented as squares, whose size indicates the weight for the meta-analytic pooled effect estimate; black lines represent 95% confidence intervals of the aggregated effect sizes within each study (Level 3); grey lines represent 95% confidence intervals of the median precision of each individual effect size (Level 2) within each study. The black diamond represents the pooled effect estimate with its 95% confidence interval. *J* is the number of biomarkers within each study.

**Figure 5 ijms-24-04445-f005:**
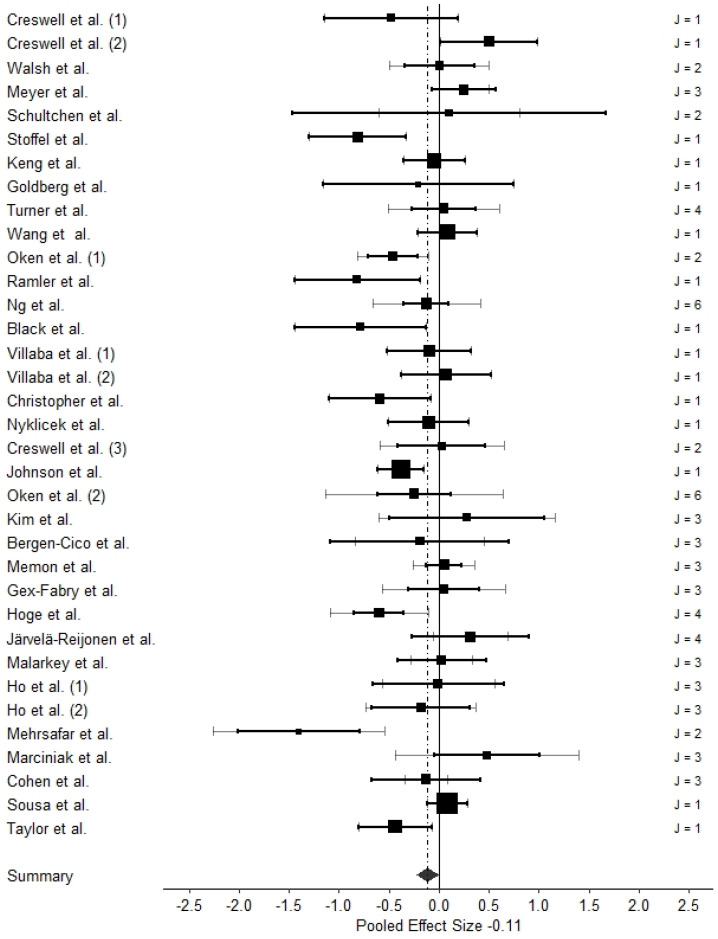
Three-Level Forest Plot of TLMA 2. Note. References of included studies are provided in Table 2. Numbers in parentheses refer to different samples of the respective studies (see Table 2 for details). Aggregated effect sizes on the study level (Level 3) are represented as squares, whose size indicates the weight for the meta-analytic pooled effect estimate; black lines represent 95% confidence intervals of the aggregated effect sizes within each study (Level 3); grey lines represent 95% confidence intervals of the median precision of each individual effect size (Level 2) within each study. The black diamond represents the pooled effect estimate with its 95% confidence interval. *J* is the number of biomarkers within each study.

**Figure 6 ijms-24-04445-f006:**
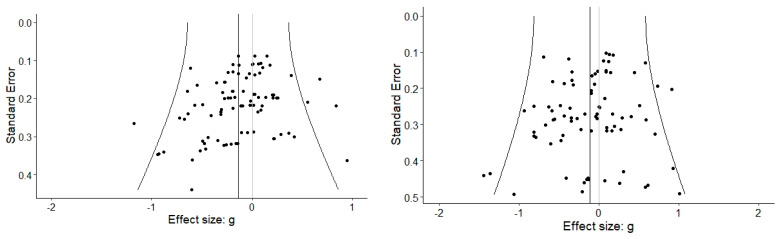
Three-Level Funnel Plots of TLMA 1 (**Left**) and TLMA 2 (**Right**). Note. The black dots represent effect sizes of individual biomarkers (Level 2), plotted against their standard errors. Asymmetry indicates possible publication bias.

**Table 1 ijms-24-04445-t001:** Biomarkers of Inflammation and Stress Accounted for in the Meta-Analyses.

Biomarker	Description	Beneficial Levels
*Biomarkers of stress*		
AA	Alpha amylase; stress-sensitive enzyme; levels change in response to physiological and psychological stress; elevated levels indicate stress [20]	↓
ACTH	Adrenocorticotropic hormone; stress-sensitive hormone; promotes cortisol production; elevated levels indicate stress [21]	↓
Cortisol	Stress-sensitive hormone; elevated levels indicate physiological and psychological stress; mediates effects of psychological distress on physiological health [22]	↓
DHEA-S	Dehydroepiandrosterone sulfate; stress-sensitive hormone; acts as cortisol antagonist; lower levels indicate stress [23]	↑
*Biomarkers of inflammation*		
CRP	C-reactive protein; inflammation-sensitive protein; elevated levels indicate inflammation; relevant for physical and mental illnesses [18]	↓
IL-6	Interleukin-6; pro-inflammatory cytokine; elevated levels indicate inflammation and are related to worsening health status; promotes sickness behavior; relevant for mental illnesses, e.g., depression [19,24]	↓
IL-8	Interleukin-8, pro-inflammatory cytokine; elevated levels indicate inflammation and are related to worsening health status; involved in pathogenesis of mental disorders [25]	↓
IL-1β	Interleukin-1β; pro-inflammatory cytokine; elevated levels indicate inflammation and are related to worsening health status; relevant for neurodegenerative diseases [26]	↓
IL-1ra	Interleukin-1ra; inflammatory cytokine; elevated levels indicate inflammation and are related to worsening health status [27]	↓
IP-10	Interferon gamma protein 10; inflammatory chemokine; elevated levels indicate inflammation and are related to worsening health status [28]	↓
NF-κB	Nuclear factor kappa-light-chain-enhancer of activated B cells; pro-inflammatory transcription factor; elevated levels indicate inflammation and are related to worsening health status; possibly concerts psychosocial stress into cellular activation [29]	↓
TNF-α	Tumor necrosis factor alpha; inflammatory cytokine; elevated levels indicate inflammation and are related to worsening health status; relevant for pathogenesis of mental disorders [19]	↓
*Biomarkers indirectly related to inflammation and stress*		
BDNF	Brain-derived neurotrophic factor; relevant for neuronal and synaptic development; reduced levels are related to neurodegenerative diseases [30]	↑
EGF	Epidermal growth factor; decreased levels seem to be relevant for the pathogenesis of stress-related and mood-related disorders [31]	↑
FKBP5m	Methylation of immune-related protein; reduced methylation negatively influences the body’s stress response, e.g., in patients with post-traumatic stress disorder [32]	↑
LTL	Leukocyte telomerase length (LTL); influenced by stress and inflammation; reduced LTL is related to mental illnesses [33]	↑
NP-Y	Neuropeptide-Y; relevant for stress and anxiety regulation; decreased levels negatively influence the body’s stress response [34]	↑
*SLC6A4* DNAm	Methylation of *SLC6A4* gene; responsible for serotonin reuptake from synaptic gap; elevated methylation leads to a faster reuptake; relevant for mood and stress-related disorders [35]	↓
sOXT	Oxytocin; seems to have anti-stress and anti-inflammatory capacities; increased levels of sOXT reduce inflammation and stress [36]	↑

**Table 2 ijms-24-04445-t002:** Primary Studies.

Authors	Year	TLMA	Area	Total *N*	Mean Age	%Female	Sample	MBI	Control	Duration	Biomarker(s)
Bellosta-Batalla et al. [80]	2020	1 + 2	EU	37	23.71	76.6	Healthy	MCBI	N/A	8	sOXT
Bergen-Cico et al. [81]	2014	1 + 2	NA	40	48	10	PTSD	PCbMP	N/A	4	Cortisol: AUCg, AUCi, CAR
Bishop et al. [82]	2018	1	NA	18	59.3	18	PTSD	MBSR	N/A	9	*SLC6A4* DNAm, FKBP5m
Black et al. [79]	2015	1 + 2	NA	49	63.3	67	At-risk	MAP	SHE *	6	NF-κB
Brand et al. [83]	2012	1	EU	11	40.2	81.81	Healthy	MBSR	N/A	8	Cortisol CAR
Christopher et al. [84]	2018	1 + 2	NA	61	43.98	10	Healthy	MBI-Resilience	N/A	8	Cortisol AUCi
Cohen et al. [85]	2021	1 + 2	NA	38	14.31	43	Stressed	MBSR-T	N/A	4	CRP, IL-6, Cortisol
Creswell et al. [86]	2012	1 + 2	NA	40	65	80	At-risk	MBSR	N/A	8	CRP, IL-6
Creswell et al. [87]	2014	1 + 2	NA	66	21.7	41	Stressed	MM	Cognitive control training *	0.5	Cortisol Reactivity
Creswell et al. [88]	2016	1 + 2	NA	35	39.47	42.85	Stressed	HEM	HER *	0.5	IL-6
Duchemin et al. [89]	2015	1	NA	16	44.2	87.5	Stressed	MBI-Workplace	N/A	8	AA
Eisendrath et al. [90]	2016	1	NA	11	34.9	72.7	Depression	MBCT	N/A	8	CRP
Fendel et al. [91]	2020	1	EU	9	33.2	55.55	Healthy	MBI-Physicians	N/A	8	Cortisol Salivary
Galantino et al. [92]	2005	1	NA	42	43	96	Healthy	MMP–Heart and Mind	N/A	8	Cortisol Salivary
Gex-Fabry et al. [93]	2012	1 + 2	EU	42	46.75	71.4	Depression	MBCT + TAU	TAU *	8	Cortisol: Slope, CAR, AUC
Goldberg et al. [94]	2014	1 + 2	SA	18	42.2	55.6	SUD	MBI-Smokers	FFS-E *	7	Cortisol Hair
Ho et al. (1) [95]	2020	1 + 2	Asia	51	6.56	39.15	Stressed	Mindful Parenting	N/A	6	Cortisol: Morning, Evening, Slope
Ho et al. (2) [95]	2020	1 + 2	Asia	51	38.75	96.15	Stressed	MM	N/A	6	Cortisol: Morning, Evening, Slope
Hoge et al. [96]	2018	1 + 2	NA	80	39	46.5	GAD	MBI	N/A	8	Cortisol AUC, ACTH, TNF-α, IL-6
Järvelä-Reijonen et al. [97]	2020	1 + 2	EU	113	49.99	84	Healthy	ACT	N/A	8	CRP, IL-1ra, Cortisol, DHEA-S
Johnson et al. [16]	2014	1 + 2	NA	281	21.55	0	Stressed	MMFT	Training as usual *	8	NP-Y
Keng et al. [98]	2020	1 + 2	SA	158	27.24	63.3	Healthy	MBSR	HEP *	8	LTL
Kim et al. [99]	2013	1 + 2	NA	22	46.3	95.45	PTSD	MBX	N/A	8	Cortisol Serum, ACTH AUC, DHEA-S
Malarkey et al. [100]	2013	1 + 2	NA	170	50	87.5	At-risk	MBI	Lifestyle education group *	8	CRP, IL-6, Cortisol
Marciniak et al. [101]	2020	1 + 2	EU	20	74	65	MCI	MBI	Cognitive training *	8	CRP, IL-6, TNF-α
Marcus et al. [102]	2003	1	NA	12	33.4	14.3	SUD	MBI	N/A	8	Cortisol AUC
Matvienko-Sikar et al. [103]	2017	1	EU	12	34.53	100	Healthy	MBI online + Gratitude	N/A	3	Cortisol: Morning, CAR, Evening
McClintock et al. [104]	2019	1	NA	72	43.4	36.1	SUD	MBRP	N/A	8	IL-6
Mehrsafar et al. [105]	2019	1 + 2	EU	26	25.4	0	Stressed	MBI -Training	N/A	8	Cortisol slope, AA slope
Memon et al. [17]	2017	1 + 2	NA	166	41.5	87.5	Depression, GAD	MBSR	CBT*	8	IL-8, CRP, EGF
Meyer et al. [76]	2019	1 + 2	NA	259	49.7	76	Healthy	MBSR	N/A	8	CRP, IL-6, IP-10
Ng et al. [106]	2020	1 + 2	Asia	55	71.28	74.6	MCI	MAP	HEP *	36	CRP, IL-1β, IL-6, Cortisol, DHEA-S, BDNF
Nyklíček et al. [107]	2013	1 + 2	EU	88	46.1	70.6	Stressed	MBSR	N/A	8	Cortisol Salivary
Oken et al. [108]	2010	1 + 2	NA	20	63.15	85	Healthy	MBI-Dementia Caregivers	N/A	7	IL-6, TNF-α, CRP, Cortisol
Oken et al. [109]	2017	1 + 2	NA	128	59.8	50	Stressed	MBI	N/A	6	Cortisol: Slope, CAR
Ramler et al. [77]	2015	1 + 2	NA	48	18	66.7	Healthy	MBI-Students	N/A	9	Cortisol Salivary
Roberts et al. [110]	2020	1	NA	47	39.38	91.5	Healthy	MBI	SME *	8	Cortisol AUCg
Schultchen et al. [111]	2019	1 + 2	EU	47	22.21	75	Stressed	MBI-Audio	Audio book *	8	DHEA-S, Cortisol Hair
de Sousa et al. [112]	2021	1 + 2	SA	40	24.15	50	Stressed	bMM	Health Education + Drawing *	0.5	Cortisol
Stoffel et al. [78]	2019	1 + 2	EU	74	21.1	64.5	Stressed	MBI-University	N/A	12	*SLC6A4* DNAm
Taylor et al. [113]	2021	1 + 2	NA	23	50	95.50	Healthy	MBI-Teachers	N/A	16	Cortisol CAR
Turner et al. [44]	2020	1 + 2	EU	53	24	70.37	Stressed	MBI-Students	Student support as usual *	8	Cortisol, IL-8, TNF-α, CRP
Villalba et al. (1) [15]	2019	1 + 2	NA	93	34.5	66.5	Stressed	MBI-MA App	My Time *	2	CRP
Villalba et al. (2) [15]	2019	1 + 2	NA	83	39.5	70.5	Stressed	MBI-Monitoring + Accepting	N/A	8	CRP
Walsh et al. [114]	2016	1 + 2	NA	64	19.13	100	Depression	MBI	Contact-control group *	4	IL-6, TNF-α
Wang et al. [79]	2017	1 + 2	EU	177	43.25	56	Depression, GAD	MBI	CBT *	8	LTL
Zgierska et al. [115]	2008	1	NA	12	38.4	50	SUD	MBRP	N/A	8	IL-6, Cortisol: CAR, Morning, Midday, Evening

Note. Area: EU = Europe; NA = North America; SA = South America. MBIs (mindfulness-based interventions): bMM = brief mindfulness meditation; HEM = health enhancement through mindfulness; MAP = mindful awareness practice; MBSR = mindfulness-base stress reduction; MBCT = mindfulness-based cognitive therapy; MBX = mindfulness-based stretching and breathing; MBRP = mindfulness-based relapse prevention; MMFT = mindfulness meditation fitness training; MMP = mindfulness meditation program; PCbMP = primary care brief mindfulness practice. Controls: CBT = cognitive behavioral therapy; FFS-E = free from smoking education; HEP = health education program; HER = health enhancement through relaxation; SHE = sleep hygiene education; SME = stress management education; TAU = treatment as usual; N/A = no control program; * = active control group. Psychiatric disorders: GAD = generalized anxiety disorder; MCI = mild cognitive impairment; PTSD = post-traumatic stress disorder; SUD = Substance use disorder. Biomarkers: AA = alpha-amylase; AUC = area under the curve; BDNF = brain-derived neurotrophic factor; CAR = cortisol awakening response; CRP = C-reactive protein; DHEA-S = dehydroepiandrosterone sulfate; EGF = epidermal growth factor; FKBP5m = methylation of binding protein 5; IL = interleukin; LTL = leukocyte telomere length; NP-Y = neuropeptide Y; *SLC6A4* DNAm = DNA methylation of the serotonin transporter gene; sOXT = salivary oxytocin; TNF-α = tumor necrosis factor alpha.

**Table 3 ijms-24-04445-t003:** Pooled Effect Size Estimates and Effect Size Heterogeneity.

Analysis	*k*	#ES	*g*	95% *CI*	*p*	τ		*I* ^2^	
						Level 2	Level 3	Level 2	Level 3
TLMA 1	42	91	−0.15	[−0.23, −0.06]	<0.001	0.20	0.18	37%	32%
Excluding outliers	40	83	−0.11	[−0.18, −0.04]	0.002	0.06	0.14	7%	36%
Including follow-up data	42	117	−0.17	[−0.25, −0.08]	<0.001	0.25	0.14	54%	16%
TLMA 2	35	79	−0.11	[−0.23, 0.001]	0.053	0.32	0.17	57%	17%
Excluding outliers	34	75	−0.15	[−0.25, −0.05]	0.003	0.26	0.13	51%	13%
Including follow-up data	35	94	−0.15	[−0.26, −0.04]	0.007	0.33	0.12	63%	9%

Note. *k* = number of independent samples; #ES = number of effect sizes; *g* = Hedges *g*; *CI* = confidence interval; τ = variability (standard deviation) of effect sizes on Levels 2 and 3; *I*^2^ = heterogeneity of effect sizes on Levels 2 and 3.

**Table 4 ijms-24-04445-t004:** Results of Subgroup Analyses in TLMA 1.

	*k*	#ES	*g*	95% *CI*	*p*	*p* of Moderator
Type of sample						0.92
Psychiatric	15	39	−0.13	[−0.27, 0.01]	0.07	
Stressed and at-risk	14	25	−0.03	[−0.24, 0.19]	0.80	
Healthy	13	27	−0.04	[−0.26, 0.17]	0.70	
Type of MBI						0.20
MBSR, MBCT, MBRP	15	31	−0.17	[−0.31, −0.03]	0.015	
Adapted MBSR	26	56	0.02	[−0.16, 0.20]	0.82	
ACT	1	4	0.39	[−0.04, 0.82]	0.08	
Duration of MBI						0.35
8 weeks	26	52	−0.12	[−0.22, −0.01]	0.031	
<8 weeks	13	30	−0.11	[−0.29, 0.08]	0.24	
>8 weeks	3	9	0.10	[−0.22, 0.43]	0.53	
Biomarker						0.18
Stress-related	27	48	−0.20	[−0.31, −0.09]	<0.001	
Inflammation-related	19	36	0.13	[−0.01, 0.28]	0.07	
Indirect	5	7	0.03	[−0.22, 0.28]	0.81	
Risk of bias						0.18
Low risk	6	15	0.03	[−0.17, 0.24]	0.76	
Some concerns	27	69	−0.21	[−0.44, 0.01]	0.07	
High risk	2	7	−0.15	[−0.55, 0.24]	0.44	

Note. *k* = number of studies; #ES = number of effect sizes; *g* = Hedges *g*; *CI* = confidence interval; stress related = AA, ACTH, cortisol, DHEA-S; inflammation related = CRP, IL-6, IL-8, IL-1β, IL-1ra, IP-10, NF-κB, TNF-α; indirect = BDNF, EGF, LTL, NP-Y, *SLC6A4* DN. The first subgroup in each moderator served as a baseline; entries for the respective other subgroups relate to deviations to this baseline.

**Table 5 ijms-24-04445-t005:** Results of Subgroup Analyses in TLMA 2.

	*K*	#ES	*g*	95% *CI*	*p*	*p* of Moderator
Type of Sample						0.65
Psychiatric	11	30	−0.09	[−0.29, 0.11]	0.40	
Stressed and at-risk	16	29	−0.09	[−0.36, 0.18]	0.51	
Healthy	8	20	0.04	[−0.27, 0.34]	0.81	
Type of MBI						0.08
MBSR, MBCT, MBRP	8	20	−0.01	[−0.22, 0.20]	0.92	
Adapted MBSR	26	55	−0.17	[−0.42, 0.08]	0.18	
ACT	1	4	0.32	[−0.20, 0.84]	0.22	
Duration of MBI						0.38
8 weeks	20	43	−0.05	[−0.21, 0.10]	0.49	
<8 weeks	12	28	−0.10	[−0.35, 0.15]	0.42	
>8 weeks	3	8	−0.27	[−0.68, 0.14]	0.20	
Biomarker						0.20
Stress-related	22	40	−0.19	[−0.34, −0.04]	0.017	
Inflammation-related	15	33	0.18	[−0.03, 0.39]	0.09	
Indirect	6	6	−0.001	[−0.36, 0.35]	0.99	
Risk of bias						0.50
Low risk	6	15	0.004	[−0.27, 0.28]	0.97	
Some concerns	26	59	−0.13	[−0.44, 0.18]	0.41	
High risk	3	5	−0.29	[−0.80, 0.22]	0.26	
Control group						0.44
Active control	18	42	−0.07	[−0.23, 0.09]	0.37	
Passive control	17	38	−0.09	[−0.33, 0.14]	0.44	

Note. *k* = number of studies; #ES = number of effect sizes; *g* = Hedges *g*; *CI* = confidence interval; stress related = AA, ACTH, cortisol, DHEA-S; inflammation related = CRP, IL-6, IL-8, IL-1β, IL-1ra, IP-10, NF-κB, TNF-α; indirect = BDNF, EGF, LTL, NP-Y, *SLC6A4* DNAm. The first subgroup in each moderator served as a baseline; entries for the respective other subgroups relate to deviations to this baseline.

## Data Availability

A protocol for this study was preregistered at OSF (https://osf.io/8h53c (registered on 27 March 2021)). All materials, analysis data, and R code are available there as well.

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
