# Peer review of "Only Small Effects of Mindfulness-Based Interventions on Biomarker Levels of Inflammation and Stress: A Preregistered Systematic Review and Two Three-Level Meta-Analyses"

_ijms, 2023, doi:10.3390/ijms24054445_

Round 1

Reviewer 1 Report

In this paper, author performed two three-level meta-analysis to comprehensive investigate the effects of MBIs on a broad variety of biomarkers related to inflammation and stress in both psychiatric and subclinical populations; thus supporting the role of  MBIs in positively effecting biomarkers.

The paper is well written, statistical analysis are adequate and complete. Authors use an innovative approach like three-level meta-analysis. Conclusions are in accordance with results obtained.

Author Response

Thank you for your positive review of our work!

Reviewer 2 Report

This is a very important study to the field of behavioural psychosocial interventions in healthcare and to the field of mind-body research. In particular with relevance to the field of MBIs and CBTs in healthcare. The study also has some very important implications concerning design future studies  - including for choice of biomarkers for stress and inflammation (and other mind-body responses).

Some minor points to address are:

1) font type/size varies in the beginning of the manuscript - but check this throughout ms.

2) lines 54-55 "...seem to be relevant for different psychiatric symptoms" ..BE MORE SPECIFIC ABOUT WHAT MAY BE IMPLICATED BY "RELEVANT"....."Favorable levels depend on...." BE MORE SPECIFIC AND PRECISE ABOUT WHATR IS MEANT BE "FAVORABLE"

3)Table 4. effect size (g=-0.07) for inflammation related biomarker is not within 95% CI - CHECK AND EXPLAIN THIS!

4) TABLE 5. Same question: g=-0.002 not within 95% CI -CHECK AND EXPLAIN THIS!

Author Response

This is a very important study to the field of behavioural psychosocial interventions in healthcare and to the field of mind-body research. In particular with relevance to the field of MBIs and CBTs in healthcare. The study also has some very important implications concerning design future studies  - including for choice of biomarkers for stress and inflammation (and other mind-body responses).

Some minor points to address are:

1) font type/size varies in the beginning of the manuscript - but check this throughout ms.

Response #1: Thank you – font type/size has been corrected throughout in the revised ms.

2) lines 54-55 "...seem to be relevant for different psychiatric symptoms" ..BE MORE SPECIFIC ABOUT WHAT MAY BE IMPLICATED BY "RELEVANT"....."Favorable levels depend on...." BE MORE SPECIFIC AND PRECISE ABOUT WHATR IS MEANT BE "FAVORABLE"

Response #2: We have changed our wording here in the revised ms. – we now write: “In the field of mental illness, elevated levels of certain biomarkers, such as C-reactive proteins (Villalba et al., 2019), neuropeptides (Johnson et al., 2014), cytokines (Memon et al., 2017), and chemokines (Köhler et al., 2017), are associated with various psychiatric symptoms (see also Dowlati et al., 2010). The direction of association may differ for other biomarkers (for details and comprehensive overview, see Table 1).”

3)Table 4. effect size (g=-0.07) for inflammation related biomarker is not within 95% CI - CHECK AND EXPLAIN THIS!

4) TABLE 5. Same question: g=-0.002 not within 95% CI -CHECK AND EXPLAIN THIS!

Response #3: We apologize for the typos in these tables. Confidence intervals for the point estimates of subgroups, which were used as baselines in these comparisons, and of deviations to these baselines in the other subgroups were mixed up. to All errors were corrected in the revised ms.

Reviewer 3 Report

The manuscript "Only Small Effects of Mindfulness-Based Interventions on Biomarker Levels of Inflammation and Stress: A Preregistered Systematic Review and Two Three-Level Meta-Analyses" presents the results of a meta-analysis addressed the effects of MBIs on biomarkers in psychiatric populations and among healthy, stressed, and at-risk populations. The study was designed appropriately to test the hypothesis. Moreover, as preregistered at OSF, the study design is well documented (PRISMA flow diagram, table of primary studies, collection of comprehensive data in supplementary materials). Thus, based on the details in the methods section, the results can be reproducible. The cited references are relevant to the topic. The manuscript is scientifically sound, with conclusions consistent with the evidence presented.

The Reviewer found some concerns regarding the following:

1.       Line 55-56 "Favourable levels depend on the unique mechanism of action of each biomarker"

Biomarker is a biological molecule found in blood, other body fluids, or tissues that is a sign of a normal or abnormal process or a condition or disease. A biomarker may be used to see how well the body responds to a treatment for a disease or condition. It seems odd to claim that biomarker has "a unique mechanism of action." Please change that sentence.

2.       It is better to put the explanations of symbols in mathematical formulas under the formula. 

Author Response

The manuscript "Only Small Effects of Mindfulness-Based Interventions on Biomarker Levels of Inflammation and Stress: A Preregistered Systematic Review and Two Three-Level Meta-Analyses" presents the results of a meta-analysis addressed the effects of MBIs on biomarkers in psychiatric populations and among healthy, stressed, and at-risk populations. The study was designed appropriately to test the hypothesis. Moreover, as preregistered at OSF, the study design is well documented (PRISMA flow diagram, table of primary studies, collection of comprehensive data in supplementary materials). Thus, based on the details in the methods section, the results can be reproducible. The cited references are relevant to the topic. The manuscript is scientifically sound, with conclusions consistent with the evidence presented.

Response #1: Thank you for this positive evaluation of our work!

The Reviewer found some concerns regarding the following:

  1. Line 55-56 "Favourable levels depend on the unique mechanism of action of each biomarker"

Biomarker is a biological molecule found in blood, other body fluids, or tissues that is a sign of a normal or abnormal process or a condition or disease. A biomarker may be used to see how well the body responds to a treatment for a disease or condition. It seems odd to claim that biomarker has "a unique mechanism of action." Please change that sentence.

Response #2: Thank you. We have changed this sentence in the revised ms. It now reads: “The direction of association may differ for other biomarkers (for details and comprehensive overview, see Table 1).”

  1. It is better to put the explanations of symbols in mathematical formulas under the formula.

Response #3: This was done in the revised ms.